# Joint Attention-Driven Domain Fusion and Noise-Tolerant Learning for Multi-Source Domain Adaptation

## Abstract

Multi-source Unsupervised Domain Adaptation (MUDA) transfers knowledge from multiple source domains with labeled data to an unlabeled target domain. Recently, endeavours have been made in establishing connections among different domains to enable feature interaction. However, as these approaches essentially enhance category information, they lack the transfer of the domain-specific information. Moreover, few research has explored the connection between pseudo-label generation and the framework's learning capabilities, crucial for ensuring robust MUDA. In this paper, we propose a novel framework, which significantly reduces the domain discrepancy and demonstrates new state-of-the-art performance. In particular, we first propose a Contrary Attention-based Domain Merge (CADM) module to enable the interaction among the features so as to achieve the mixture of domain-specific information instead of focusing on the category information. Secondly, to enable the network to correct the pseudo labels during training, we propose an adaptive and reverse cross-entropy loss, which can adaptively impose constraints on the pseudo-label generation process. We conduct experiments on four benchmark datasets, showing that our approach can efficiently fuse all domains for MUDA while showing much better performance than the prior methods.

## 1 Introduction

Deep neural networks (DNNs) have achieved excellent performance on various vision tasks under the assumption that training and test data come from the same distribution. However, different scenes have different illumination, viewing angles, and styles, which may cause the domain shift problem (Zhu et al., 2019; Tzeng et al., 2017; Long et al., 2016). This can eventually lead to a significant performance drop on the target task.

Unsupervised Domain Adaptation (UDA) aims at addressing this issue by transferring knowledge from the source domain to the unlabeled target domain (Saenko et al., 2010). Early research has mostly focused on Single-source Unsupervised Domain Adaptation (SUDA), which transfers knowledge from one source domain to the target domain. Accordingly, some methods align the feature distribution among source and target domains (Tzeng et al., 2014) while some (Tzeng et al., 2017) learn domain invariants through adversarial learning. Liang et al. (2020) use the label information to maintain the robust training process. However, data is usually collected from multiple domains in the real-world scenario, which arises a more practical task, i.e., Multi-source Unsupervised Domain Adaptation (MUDA) (Duan et al., 2012).

MUDA leverages all of the available data and thus enables performance gains; nonetheless, it introduces a new challenge of reducing domain shift between all source and target domains. For this, some research (Peng et al., 2019) builds their methods based on SUDA, aiming to extract common domain-invariant features for all domains. Moreover, some works, *e.g.*, Venkat et al. (2021); Zhou et al. (2021) focus on the classifier's predictions to achieve domain alignment. Recently, some approaches (Li et al., 2021; Wen et al., 2020) take advantage of the MUDA property to create connections for each domain. Overall, since the main challenge of MUDA is to eliminate the differences between all domains, there are two main ways to achieve this. One is to extract domain invariant features among all domains, i.e., filter domain-specific information for different domains. The other is by mixing domain-specific information from different domains so that all domains share such

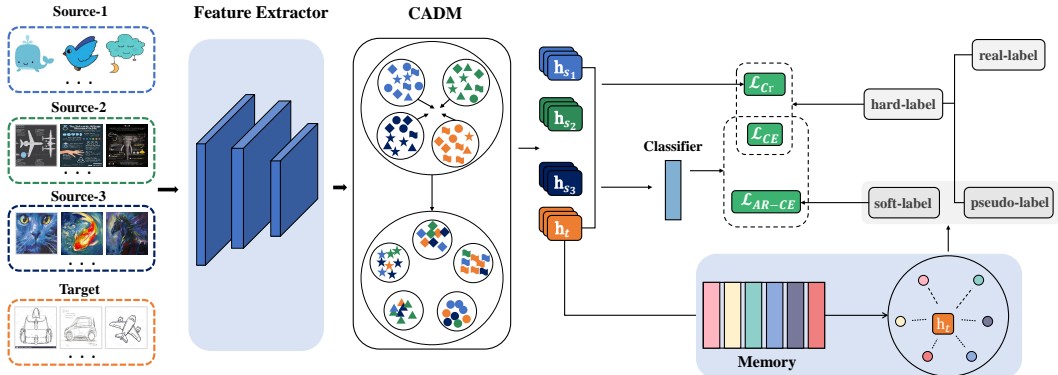

Figure 1: Overview of our proposed framework that consists of three primary elements: (1) The feature extractor extracts features from various domains. (2) CADM is proposed to implement message passing and fuse features from different domains. The same domain is represented by the same color, while different shapes represent different classes. (3) we propose AR-CE loss and use the maintained memory to compute the soft label and pseudo label (hard label) for target domain.

mixed information and thus **fuse into one domain**. Previous approaches have mostly followed the former, however, filtering the domain-specific information for multiple domains can be difficult and often results in losing discrimination ability. For the latter, few methods have been proposed to address MUDA in this way, and there is a lack of effective frameworks to achieve such domain fusion, which is the main problem to be addressed by our proposed approach. Moreover, existing methods ignore the importance of generating reliable pseudo labels as noisy pseudo labels could lead to the accumulation of prediction errors. Consequently, it is imperative to design a robust pseudo-label generation based on the MUDA framework.

In this paper, we propose a novel framework that better reduces the domain discrepancy and shows the new state-of-the-art (SoTA) MUDA performance, as shown in Fig. 1. Our method enjoys two pivotal technical breakthroughs. Firstly, we propose a Contrary Attention-based Domain Merge (CADM) module (Sec. 4.2), whose role is to perform the domain fusion. Self attention (Vaswani et al., 2017; Dosovitskiy et al., 2020) can capture the higher-order correlation between features and emphases more relevant feature information, *e.g.*, the semantically closest information. Differently, our CADM proposes the contrary attention, enabling each domain to pay more attention to semantically different domain-specific information of other domains. Then, by integrating these domain-specific information, each domain can achieve movement to other domains, thus resulting in domain fusion. Secondly, to enable the network to correct the pseudo labels during training, we take the pseudo-label generation as the optimization objective of the network by proposing an adaptive and reverse cross-entropy (AR-CE) loss (Sec. 4.3). It imposes the optimizable constraints for pseudo-label generation, enabling the network to correct pseudo labels that tend to be wrong and reinforce pseudo labels that tend to be correct.

We conduct extensive experiments on four benchmark datasets. The results show our method achieves new state-of-the-art performance and especially has significant advantages in dealing with the large-scale dataset. In summary, we have made the following contributions:

- We propose a CADM to fuse the features of source and target domains, bridging the gap between different domains and enhancing the discriminability of different categories.

- We propose a loss function, AR-CE loss, to diminish the negative impact of the noisy pseudo label during training.

- Our method demonstrates the new state-of-the-art (SoTA) MUDA performance on multiple benchmark datasets.

## 2 RELATED WORK

### 2.1 UNSUPERVISED DOMAIN ADAPTATION (UDA)

According to the number of source domains, UDA can be divided into Single-source Unsupervised Domain Adaptation (SUDA) and Multi-source Unsupervised Domain Adaptation (MUDA). In SUDA, some methods use the metrics, such as Maximum Mean Discrepancy (Long et al., 2017; Wang et al., 2018), to define and optimize domain shift. Some methods (Sankaranarayanan et al., 2018; Ma et al., 2019) learn the domain-invariant representations through adversarial learning. Other methods are based on the label information. Liang et al. (2020) aligns domains implicitly by constraining the distribution of prediction. Following Liang et al. (2020), Liang et al. (2021) introduces a label shifting strategy as a way to improve the accuracy of low confidence predictions. For MUDA, previous methods eliminate the domain shift while aligning all source domains by constructing multiple source-target pairs, where (Peng et al., 2019; Zhu et al., 2019) are based on the discrepancy metric among different distributions and (Venkat et al., 2021; Ren et al., 2022) are based on the adversarial learning. Some methods (Venkat et al., 2021; Nguyen et al., 2021a; Li et al., 2022; Nguyen et al., 2021b) explore the relation between different classifiers and develop different agreements to achieve domain alignment. Some (Wen et al., 2020; Ahmed et al., 2021) achieve an adaptive transfer by a weighted combination of source domains. Current methods (Wang et al., 2020; Li et al., 2021) mainly construct connections between different domains to enable feature interaction and explore the relation of category information from different domains. Our approach also constructs connections between different domains; however, it mainly transfers domain-specific information to achieve the fusion of all domains.

### 2.2 ROBUST LOSS FUNCTION UNDER NOISY LABELS

A robust loss function is critical for UDA because the unlabeled target domain requires pseudo labels, which are often noisy. Previous work (Ghosh et al., 2017) demonstrates that some loss functions such as Mean Absolute Error (MAE) are more robust to noisy labels than the commonly used loss functions such as Cross Entropy (CE) loss. (Wang et al., 2019b) proposes the Symmetric Cross Entropy (SCE) combining the Reverse Cross Entropy (RCE) with the CE loss. Moreover, (Wang et al., 2019a) shows that directly adjusting the update process of the loss by the weight variance is effective. Some methods (Venkat et al., 2021; Zhou et al., 2021) mentioned in Section 2.1 focus on exploring a learning strategy to handle prediction values that have high or low confidence levels, respectively. Differently, we design an adaptive loss function for the network to learn robust pseudo-label generation by self-correction.

## 3 PROBLEM SETTING

For MUDA task, there are $N$ source distributions and one target distribution, which can be denoted as $\{p_{s_j}(x,y)\}_{j=1}^N$ and $\{p_t(x,y)\}$. The labeled source domain images $\{X_{s_j}, Y_{s_j}\}_{j=1}^N$ are obtained from the source distributions, where $\mathcal{X}_{s_j} = \{x_{s_j}^i\}_{i=1}^{|\mathcal{X}_{s_j}|}$ are the images in the source domain $j$ and $\mathcal{Y}_{s_j} = \{y_{s_j}^i\}_{i=1}^{|\mathcal{X}_{s_j}|}$ represents the corresponding ground-truth labels. As for the unlabeled data in the target domain, there are target images $\mathcal{X}_t = \{x_t^i\}_{i=1}^{|\mathcal{X}_t|}$ from target distribution. In this paper, we uniformly set the number of samples in a batch as $B$, with $b = \frac{B}{N+1}$ for every domain (including the target domain). For a single sample $x_{s_j}^i$, the subscript $s_j$ represents the $j$-th source domain, while the superscript $i$ indicates that it is the $i$-th sample in that domain.

## 4 PROPOSED METHOD

### 4.1 OVERALL FRAMEWORK

The general MUDA pipeline consists of two phases: 1) pre-training on the source domain; 2) training when transferring to the target domain. The base structure of the model consists of a feature extractor $F(\cdot)$ and a classifier $C(\cdot)$. Previous approaches mostly use different extractors or classifiers, unlike

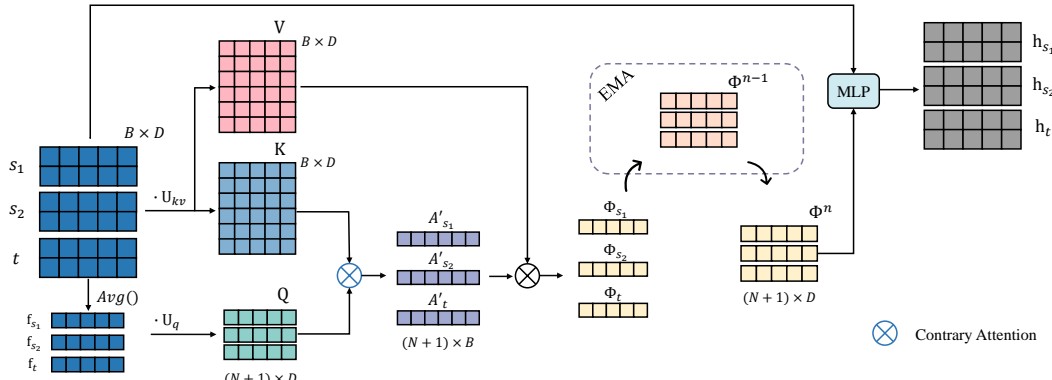

Figure 2: CADM review: Features from different domains are fused by CADM. For simplicity, we set $N$, $B$, and $D$ as 2, 6, and 5, respectively. Domain prototypes are mapped to $\mathbf{Q}$, while all features are mapped to $\mathbf{K}$ and $\mathbf{V}$. The domain style centroids are obtained by the contrary attention map $A'$ and $\mathbf{V}$, which is then updated by EMA. Finally, the fused feature $\mathbf{h}$ is obtained by MLP.

them, our approach expects to achieve a deep fusion of all domains, i.e., belonging to one domain, and therefore $F$ and $C$ are shared among domains. In the pre-training phase, we use the source domain data $\{X_{s_j}, Y_{s_j}\}_{j=1}^N$ for training to obtain the pre-trained $F(\cdot)$ and $C(\cdot)$. In the training phase for the target domain, we firstly feed the source domain data $\{X_{s_j}, Y_{s_j}\}_{j=1}^N$ and target domain data $X_t$ into $F(\cdot)$ to get the features $\mathbf{f}$, and perform the feature fusion by CADM to get the fused feature $\mathbf{h}$. Then, we use the fused features of the target domain for the pseudo-label generation and feed the fused features of all domains into $C(\cdot)$ to get prediction values. Finally, we compute the loss based on the labels. Since source training is general, training in this paper typically refers to target domain training.

## 4.2 CONTRARY ATTENTION-BASED DOMAIN MERGE MODULE

The proposed CADM enables the passing of domain-specific information to achieve the deep fusion of all domains. For each domain, we expect it can receive domain-specific information from other domains by the information transfer to move towards other domains as shown in Fig. 1. To achieve this goal, the transfer of semantically distinct domain-specific information should be encouraged. It is because receiving semantically close domain-specific information primarily strengthens the original domain information and does not help eliminate the domain shift. Instead, receiving different domain-specific information can contribute to the mixing of domain-specific information and 'push' the domain to other domains, thus achieving domain fusion. As a result, we construct contrary attention to focus on semantic differences among domains. The specific structure is shown in Fig. 2. Specifically, features $\mathbf{f}$ are first extracted from the samples via $F$, and then be mapped into the latent space to obtain query, key, and value feature:

$$\mathbf{Q} = [\mathbf{f}_{s_1} \dots \mathbf{f}_{s_N}, \mathbf{f}_t] \mathbf{U}_q \qquad \mathbf{U}_q \in \mathbb{R}^{D \times D}, \tag{1}$$

$$[\mathbf{K}, \mathbf{V}] = [\mathbf{f}_{s_1}^1 \dots \mathbf{f}_{s_2}^1 \dots \mathbf{f}_t^b] \mathbf{U}_{kv} \qquad \mathbf{U}_{kv} \in \mathbb{R}^{D \times 2D}, \tag{2}$$

where $\mathbf{U}_q$ and $\mathbf{U}_{kv}$ are learnable projection matrices, $D$ is the feature dimension, $\mathbf{f}_{s_1}^1$ is the feature obtained from the sample $x_{s_1}^1$, and $\mathbf{f}_{s_1}$ is the domain prototype representing the domain information of $s_1$. Let $g$ represent any domain, the domain prototype is obtained as follows:

$$\mathbf{f}_g = \frac{1}{b} \sum_{i=1}^b \mathbf{f}_g^i. \tag{3}$$

Since the query of each domain and the key of all samples are obtained in Eq. 1,2, then we use the query feature $\mathbf{q}_g$ in $\mathbf{Q}$ to compute the correlation with the samples of different domains:

$$A_g = \text{Softmax}\left(\frac{\mathbf{q}_g \cdot \mathbf{K}^{\mathrm{T}}}{\sqrt{D}}\right). \qquad A_g \in \mathbb{R}^{1 \times B} \tag{4}$$

We use the self-attention mechanism to model the correlation because of its powerful ability to capture dependencies, and $A_g$ is the obtained attention map. The value in $A_g$ reflects the correlation between the domain $g$ and the samples from other domains, while, as mentioned before, it is more meaningful to focus on semantically different domain-specific information. Consequently, we 'reverse' the obtained $A_g$ to generate a new attention map, which we refer to as the contrary attention map:

$$A_g' = \frac{\mathbf{1} - A_g}{\text{sum}(\mathbf{1} - A_g)}, \qquad A_g' \in \mathbb{R}^{1 \times B} \tag{5}$$

where $\mathbf{1}$ is an all-one vector. The $A_g'$ thus obtained can achieve the opposite effect of the previous $A_g$, where those features that are more different from the domain $g$ in terms of domain-specific information are assigned higher weights.

Subsequently, we use $A_g'$ to perform the feature fusion for domain $g$, which allows the merging of information from features according to their degree of dissimilarity with domain $g$:

$$\begin{aligned} \Phi_g &= A_g' \cdot \mathbf{V} \\ &= \sum_{\hat{g} \in \mathcal{G}} \sum_{i=1}^{b} a'_{g\hat{g}^i} \mathbf{v}_g^i, \end{aligned} \tag{6}$$

where $a'_{g\hat{g}^i}$ represents the contrary attention weight between the prototype of $g$ and the $i$-th feature in $\hat{g}$. $\Phi_g$ is obtained by weighting features from different domains and summing them according to their discrepancies from $g$. Given its greater emphasis on the merging of knowledge with distinct styles, we call $\Phi_g$ the domain style centroid with respect to $g$. In addition, to guarantee the robustness of $\Phi_g$, we use exponential moving averages (EMA) to maintain it:

$$(\bar{\Phi}_g)^n = (1 - \alpha)(\bar{\Phi}_g)^{n-1} + \alpha(\Phi_g)^n, \tag{7}$$

where $n$ denotes the $n$-th mini-batch. The recorded $\bar{\Phi}_g$ will be used in the inference phase.

Finally, we integrate the domain style centroid $\bar{\Phi}_g$ into the original features. A fundamental MLP is used to simulate this process:

$$\mathbf{h}_g^i = \text{MLP}\left([\mathbf{f}_g^i, \bar{\Phi}_g]\right), \tag{8}$$

where $\mathbf{h}_g^i$ is the fused feature and $\bar{\Phi}_g$ is integrated into every feature in $g$, thus creating the overall movement of domain. Through the process described above, the original features achieve the merging of different domain-specific information by integrating the domain style centroid. Meanwhile, to maintain a well-defined decision boundary during such domain fusion, we use a metric-based loss (Wen et al., 2016) to optimize the intra-class distance. It allows samples from the same categories to be close to each other and shrink around the class center.

$$\mathcal{L}_{Cr} = \sum_{g \in \mathcal{G}} \sum_{i=1}^{b} \|\mathbf{h}_g^i - \mu_{y_g^i}\|_2^2, \tag{9}$$

where $y_g^i$ is the label of $\mathbf{h}_g^i$ and $\mu_{y_g}^i \in \mathbb{R}^{1 \times D}$ denotes the $y_g^i$-th class center. For the robustness of class center, we maintain a memory M to store features at the class level and update them batch by batch during training, which is widely applied in UDA (Saito et al., 2020). The class-level features can be regarded as the class center, and we then apply them in the computation of $\mathcal{L}_{Cr}$. Specifically, we traverse all the datasets to extract features and obtain different class centers before the training process. In the target domain, we use the classification result as the category since there is no ground-truth label:

$$\mu_k = \text{Avg}\Big(\sum_{j=1}^{N} \sum_{x_i \in \mathcal{X}_{s_j}}^{y_i = k} \mathbf{h}_{s_j}^i + \sum_{x_i \in \mathcal{X}_t}^{\arg\max_j p_j(x_i) = k} \mathbf{h}_t^i\Big), \tag{10}$$

where $p_j(x)$ represents the confidence level of predicting $x$ as the $j$-th class. Then after each back-propagation step, we update the memory M with the following rule:

$$\mu_k^{new} = (1 - \beta)\mu_k^{old} + \beta\mathbf{h}, \tag{11}$$

where $k$ is the label of $\mathbf{h}$ and $\mu_k$ is the $k$-th class center in M. Under the supervision of $\mathcal{L}_{Cr}$, the features are able to maintain intra-class compactness in domain movement.

Overall, through CADM, we establish connections among domains and enable the deep fusion of different domains. Eventually, with the $\mathcal{L}_{Cr}$, the model is able to reach a tradeoff between domain merging and a well-defined decision boundary.

### 4.3 ADAPTIVE AND REVERSE CROSS ENTROPY LOSS

General UDA methods use the K-means-like algorithm to generate pseudo labels for the entire unlabeled target dataset. However, it has been observed that the pseudo labels generated in this way are often misassigned due to the lack of constraints, which introduces noise and impairs performance. To address this issue, we creatively propose to assign gradients to the generation process of pseudo labels and constrain it with a well-designed rule, so that the network can optimize both the prediction values and pseudo labels simultaneously. To begin, we expect to obtain the pseudo labels with gradient flow. Since the dynamically updated memory $\mathrm{M}$ is introduced in Section 4.2, we can obtain both soft labels and hard labels by computing the distance between features and the class-level features in $\mathrm{M}$ in every batch:

$$q(y = k|x) = \frac{e^{\mathbf{h} \cdot \mu_k^{\mathrm{T}}}}{\sum_{j=1}^{K} e^{\mathbf{h} \cdot \mu_j^{\mathrm{T}}}}, \tag{12}$$

$$\hat{y} = \arg\max_k q(y = k|x), \tag{13}$$

where $q(y|x)$ is the soft label and $\hat{y}$ is the hard label. The soft label makes the gradient backpropagation possible. Specifically, previous experimental observations (Venkat et al., 2021; Wang et al., 2019b) demonstrate that the trained classifier has important implications, which is key to preventing the model from collapsing in the unlabeled target domain. Therefore, based on the output of classifier, we establish constraints on the pseudo label by a reverse cross entropy loss:

$$\ell_{rce}(x) = \sum_{k=1}^{K} p_k(x) \log q(y = k|x), \tag{14}$$

where $k$ represents the $k$-th class, and $p_k(x)$ represents the classifier's output at the $k$-th class. As shown in the Eq. 14, unlike the cross entropy loss, which uses labels to supervise the output of the classifier, $\ell_{rce}$ uses the output of the classifier to supervise the generated soft labels. Then the gradient can be back-propagated through the soft label, thus preventing the accumulation of errors from pseudo-label generation. However, it is unfair to apply the same rule to those pseudo labels that tend to be accurate. An ideal situation is that the loss can be adaptively adjusted to correspond to different pseudo labels. Therefore, we propose the adaptive factor $a(x)$ to achieve the adaptive adjustment of $\ell_{rce}$, which is denoted as:

$$\delta_j(x, \hat{y}) = \frac{p_j(x)}{1 - p_{\hat{y}}(x)}, \tag{15}$$

$$a(x) = \sum_{j}^{j \neq \hat{y}} \delta_j(x, \hat{y}) \log \delta_j(x, \hat{y}), \tag{16}$$

where $\hat{y}$ is the generated hard label of the target domain sample $x$. We consider the classifier's output other than the class $\hat{y}$ as a distribution and the probability of $j$-th class is denoted as $\delta_j(x, \hat{y})$ in Eq.15. The entropy of this distribution then serves as the adaptive factor $a(x)$. It is because that pseudo label essentially arises from the feature extracted by the network, so its accuracy is closely related to the ability of the network to distinguish this sample. Liang et al. (2021) demonstrates that when the model has a good distinguishing ability for one sample, the output of classifier should be close to the one-hot vector. Consequently, the value of $a(x)$, in this case, is larger than the case of poor discriminatory ability. Based on $a(x)$ and $\ell_{rce}$, we design an adaptive and reverse cross entropy loss:

$$\ell_{arce}(x, \hat{y}) = \frac{\ell_{rce}(x)}{\exp(a(x)) / \tau}, \tag{17}$$

$$\mathcal{L}_{AR-CE} = \sum_{i}^{B} \ell_{arce}(x_i, \hat{y}_i). \tag{18}$$

The exponential form and $\tau$ are used to amplify its effect. Since $\mathcal{L}_{AR-CE}$ aims to focus on the optimization of pseudo label, we freeze the gradient of the prediction values in Eq. 14. The back-propagation of $\mathcal{L}_{AR-CE}$ can eventually enable the optimization of pseudo-label generation.

Furthermore, the cross entropy loss is denoted as follows:

$$\mathcal{L}_{CE} = -\frac{1}{B} \sum_{i=1}^{B} \log p_{y_i}(x_i), \tag{19}$$

where $y_i$ is the ground-truth label for source domain sample and is the hard pseudo label $\hat{y}_i$ obtained by Eq. 13 when $x_i$ is in the target domain.

Ultimately, the loss function for the entire framework can be represented as follows:

$$\mathcal{L} = \mathcal{L}_{CE} + \mathcal{L}_{Cr} + \mathcal{L}_{AR-CE}, \tag{20}$$

where $\mathcal{L}_{CE}$ is the normal cross entropy loss, $\mathcal{L}_{Cr}$ is for feature fusion, $\mathcal{L}_{AR-CE}$ is used for noise tolerant learning, and the latter one item acts only on the target domain.

## 5 EXPERIMENTS

### 5.1 DATASETS AND IMPLEMENTATION

Office-31 (Saenko et al., 2010) is a fundamental domain adaptation dataset comprised of three separate domains: Amazon (A), Webcam (W), and DSLR (D). Office-Caltech (Gong et al., 2012) has an additional domain Caltech-256 (C) than Office-31. Office-Home (Venkateswara et al., 2017) is a medium-size dataset which contains four domains: Art (Ar), Clipart (Cl), Product (Pr), and Real-World (Rw). DomainNet (Peng et al., 2019) is the largest dataset available for MUDA. It contains 6 different domains: Clipart (Clp), Infograph (Inf), Painting (Pnt), Quickdraw (Qdr), Real (Rel), and Sketch (Skt).

For a comparison with our method, we introduce the SoTA methods on MUDA, including DCTN (Xu et al., 2018), M³SDA (Peng et al., 2019), MFSAN (Zhu et al., 2019), MDDA (Zhao et al., 2020), DARN (Wen et al., 2020), Ltc-MSDA (Wang et al., 2020), SHOT++ (Liang et al., 2021), SImpAl (Venkat et al., 2021), T-SVDNet (Li et al., 2021), CMSS (Ahmed et al., 2021), DECISION (Ahmed et al., 2021), DAEL (Zhou et al., 2021), PTMDA (Ren et al., 2022) and DINE (Liang et al., 2022). For Office-31, Office-Caltech and Office-Home, we use an Imagenet (Deng et al., 2009) pre-trained ResNet-50 (He et al., 2016) as the feature extractor. As for the DomainNet, we use ResNet-101 instead. In addition, a bottleneck layer (consists of two fully-connected layers) is used as the classifier. We use the Adam optimizer with a learning rate of $1 \times 10^{-6}$ and a weight decay of $5 \times 10^{-4}$. We set the learning rate of extractor to ten times that of the classifier. The hyperparameters $\alpha$ and $\beta$ are set to 0.05 and 0.005, respectively. All the framework is based on the Pytorch (Paszke et al., 2019).

### 5.2 RESULTS

We compare the ADNT with different approaches on four datasets, and the results are shown in the following tables. Note that the one pointed by the arrow is the target domain and the others are the source domains. As shown in Table 1, the experiments on Office-31 show optimal results. In particular, it is second only to PTMDA in the ResNet50-based approach and achieves the best results in all scenarios, except for transferring W and D to A. On the Office-Caltech dataset, our method not only outperforms all current approaches, but also achieves an impressive $100\%$ accuracy on the task of transferring to W and to D. In addition, our method achieves the best results on all tasks except transfer to A, and has a significant advantage over the ResNet-101-based method. As shown in Table 3, on the medium-sized Office-Home dataset, our proposed ADNT is the best performer and has a $0.9\%$ lead over the best ResNet-50-based method. Furthermore, compared with the ResNet-101-based methods (DINE and SImpAl$_{101}$), our approach achieves leading performance with fewer parameters and a simpler model structure. As shown in Table 4, ADNT also demonstrates its strength on the largest and most challenging dataset. The total result outperforms the current optimal method by $2.3\%$, which is highly impressive on DomainNet. Overall, our approach not

Table 1: Classification accuracies (%) on Office-31 dataset, " * " indicates that the method is based on ResNet-101.

| Method | A,W →D | A,D →W | D,W →A | Avg |
|---|---|---|---|---|
| MDDA | 99.2 | 97.1 | 56.2 | 84.2 |
| LtC-MSDA | 99.6 | 97.2 | 56.9 | 84.6 |
| DCTN | 99.3 | 98.2 | 64.2 | 87.2 |
| MFSAN | 99.5 | 98.5 | 72.7 | 90.2 |
| SImpAl$_{50}$ | 99.2 | 97.4 | 70.6 | 89.0 |
| DECISION | 99.6 | 98.4 | 75.4 | 91.1 |
| DINE* | 99.2 | 98.4 | **76.8** | 91.5 |
| PTMDA | **100** | **99.6** | 75.4 | **91.7** |
| **ADNT (Ours)** | **100** | **99.6** | 74.4 | 91.3 |

Table 2: Classification accuracies (%) on Office-Caltech dataset. " * " indicates that the method is based on ResNet-101.

| Method | A,C,D →W | A,C,W →D | A,W,D →C | C,D,W →A | Avg |
|---|---|---|---|---|---|
| DCTN | 99.4 | 99.0 | 90.2 | 92.7 | 95.3 |
| M$^3$SDA | 99.4 | 99.2 | 91.5 | 94.1 | 96.1 |
| SImpAl$_{50}$ | 99.3 | 99.8 | 92.2 | 95.3 | 96.7 |
| CMSS | 99.3 | 99.6 | 96.6 | 93.7 | 97.2 |
| SHOT++ | **100** | 99.4 | 96.5 | 96.2 | 98.0 |
| DECISION | 99.6 | **100** | 95.9 | 95.9 | 98.0 |
| DINE* | 98.9 | 98.5 | 95.2 | 95.9 | 97.1 |
| PTMDA | **100** | **100** | 96.5 | **96.7** | 98.3 |
| **ADNT (Ours)** | **100** | **100** | **97.6** | 96.3 | **98.5** |

Table 3: Classification accuracies (%) on Office-Home dataset, " * " indicates that the method is based on ResNet-101.

| Method | Cl,Pr,Rw →Ar | Ar,Pr,Rw →Cl | Ar,Cl,Rw →Pr | Ar,Cl,Pr →Rw | Avg |
|---|---|---|---|---|---|
| MFSAN | 72.1 | 62.0 | 80.3 | 81.8 | 74.1 |
| SImpAl$_{50}$ | 70.8 | 56.3 | 80.2 | 81.5 | 72.2 |
| SImpAl$_{101}$* | 73.4 | 62.4 | 81.0 | 82.7 | 74.8 |
| DARN | 70.0 | **68.4** | 82.7 | 83.9 | 76.3 |
| SHOT++ | 73.1 | 61.3 | 84.3 | **84.0** | 75.7 |
| DECISION | 74.5 | 59.4 | 84.4 | 83.6 | 75.5 |
| DINE* | **74.8** | 64.1 | 85.0 | **84.6** | 77.1 |
| **ADNT (Ours)** | 73.8 | 66.5 | **85.1** | 83.3 | **77.2** |

only achieves the best results, but also has better performance on large datasets with more domains, especially DomainNet. For tasks with degraded performance, in addition to gaps in model structure (compared to ResNet-101), they mainly appear when transferring to some less stylized domains (Amazon, Clipart, Quickdraw). Further analysis is presented in the Appendix.

Table 4: Classification accuracies (%) on DomainNet dataset.

| Methods | → Clp | → Inf | → Pnt | → Qdr | → Rel | → Skt | Avg |
|---|---|---|---|---|---|---|---|
| DCTN | 48.6±0.70 | 23.5±0.60 | 48.8±0.60 | 7.2±0.50 | 53.5±0.60 | 47.3±0.50 | 38.2 |
| M$^3$SDA | 58.6±0.53 | 26.0±0.89 | 52.3±0.55 | 6.3±0.58 | 62.7±0.51 | 49.5±0.76 | 42.6 |
| MDDA | 59.4±0.60 | 23.8±0.80 | 53.2±0.60 | 12.5±0.60 | 61.8±0.50 | 48.6±0.80 | 43.2 |
| CMSS | 64.2±0.20 | 28.0±0.20 | 53.6±0.40 | 16.0±0.10 | 63.4±0.20 | 53.8±0.40 | 46.5 |
| T-SVDNet | 66.1±0.4 | 25.0±0.8 | 54.3±0.7 | **16.5±0.9** | 65.4±0.5 | 54.6±0.6 | 47.0 |
| LtC-MSDA | 63.1±0.5 | **28.7±0.7** | 56.1±0.5 | 16.3±0.5 | 66.1±0.6 | 53.8±0.6 | 47.4 |
| DAEL | **70.8±0.14** | 26.5±0.13 | 57.4±0.28 | 12.2±0.70 | 65.0±0.23 | 60.6±0.25 | 48.7 |
| PTMDA | 66.0±0.3 | 28.5±0.2 | 58.4±0.4 | 13.0±0.5 | 63.0±0.24 | 54.1±0.3 | 47.2 |
| **ADNT (ours)** | 69.0±0.38 | 28.2±0.41 | **60.5±0.45** | 16.3±0.66 | **68.7±0.63** | **63.5±0.69** | **51.0** |

## 5.3 ABLATION STUDY

To verify the effectiveness of the proposed components, we conduct ablation experiments on the Office-Home dataset, as shown in Table 5. In order to determine whether the attention-driven module actually achieves domain fusion, we conduct experiments with only the domain fusion without $\mathcal{L}_{Cr}$, which is shown as '+ CADM (w/o $\mathcal{L}_{Cr}$)' in the table. Additionally, due to the fact that our proposed contrary attention differs from general attention by paying attention to the fusion of domain-specific information, a comparison with general attention can evaluate the viability of our strategy.

Table 5: Classification accuracies (%) of ablation study on Office-Home dataset.

| Method | Cl,Pr,Rw →Ar | Ar,Pr,Rw →Cl | Ar,Cl,Rw →Pr | Ar,Cl,Pr →Rw | Avg |
|---|---|---|---|---|---|
| Baseline | 67.1 | 56.6 | 78.6 | 77.0 | 69.8 |
| + $\mathcal{L}_{R-CE}$ | 67.5 | 62.9 | 78.9 | 77.4 | 71.6 |
| + $\mathcal{L}_{AR-CE}$ | 69.2 | 63.1 | 81.2 | 78.6 | 73.0 |
| + ADM | 69.7 | 60.2 | 81.3 | 79.5 | 72.7 |
| + CADM (w/o $\mathcal{L}_{Cr}$) | 72.3 | 62.2 | 83.1 | 83.1 | 75.2 |
| + CADM (w/ $\mathcal{L}_{Cr}$) | 73.1 | 64.2 | 84.3 | 83.7 | 76.3 |
| + CADM (w/ $\mathcal{L}_{Cr}$) + $\mathcal{L}_{AR-CE}$ | 73.8 | 66.5 | 85.1 | 83.3 | 77.2 |

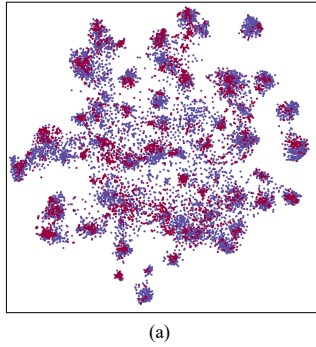 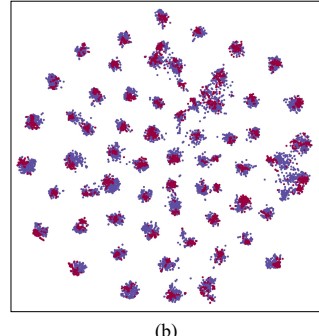 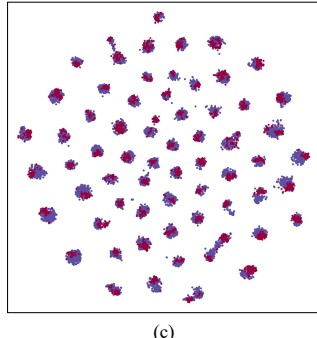

(a)               (b)               (c)

Figure 3: t-sne: The results of feature visualization on the Offce-Home dataset, while performing Ar, Cl, Rw to Pr. (a) (b) and (c) represent the result of Source-only, CADM without $\mathcal{L}_{Cr}$ and CADM with $\mathcal{L}_{Cr}$ respectively. All source domains are represented in red, while target domain is represented in blue.

Therefore, we replace the contrary attention map in the proposed CADM with a general attention map, i.e., without Eq. 5. The result of general attention is shown as '+ ADM' in Table 5. The significant performance improvement powerfully demonstrates the effectiveness of our proposed CADM. We also use t-sne (Van der Maaten & Hinton, 2008) to visualize the results with and without CADM, as shown in Fig. 3. We can observe that with the designed CADM, the different distributions are mixed with each other, achieving a higher degree of fusion. Meanwhile, the decision boundaries of different categories are more distinct.

Table 5 verifies the effectiveness of our AR-CE loss. Additionally, we conduct experiments using only reverse cross entropy loss, denoted as $+\mathcal{L}_{R-CE}$. As shown in the Table 5, $\mathcal{L}_{R-CE}$ brings the performance improvement, and the adaptive factor is able to amplify this improvement. This demonstrates the positive and effective role of adaptive factor. Due to the space limit, additional analytical experiments, including the visualization of the training process and analysis of hyperparameters, can be found in the appendix.

## 6  CONCLUSION

This paper proposes the ADNT, a framework that combines the domain fusion module and robust loss function under noisy labels. Firstly, We construct a contrary attention-based domain merge module, which can achieve a mixture of different domain-specific information and thus fuse features from different domains. In addition, we design a robust loss function to avoid the error accumulation of pseudo-label generation. Substantial experiments on four datasets have shown the superior performance of ADNT, which strongly demonstrates the effectiveness of our approach.

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

# A APPENDIX

## A.1 ADDITIONAL EXPERIMENTS ABOUT CADM

Our proposed CADM aims to achieve the domain fusion through the transfer of domain-specific information, where the coefficients in the Contrary Attention Map $A'$ have an important role to focus on the semantically different domain-specific information. To verify the action of contrary attention, we perform the visualization by Grad-CAM. Specifically, we used Grad-CAM to visualize the weights in $A'$, as the column 'Painting' in Fig. 9, reflecting the attention of other domains to that sample in Painting domain. **Since our approach emphasizes attention to domain-specific information from other domains, it is reasonable to focus on texture and background, etc., rather than content.** As shown in Fig 9, the designed contrary attention is able to focus on domain-specific information, such as texture, color, background, and other domain characteristics (*e.g.*, handwritten text in Sketch), rather than content. This makes the domain fusion possible. Meanwhile, the visualization results on domain Quickdraw show that since this domain is composed of simple lines, there is not enough information related to texture, background, etc., and the attention can be limited. This also explains why our approach does not achieve the best performance on some tasks, except for the reason of different model structure (ResNet-50 and ResNet-101). This is because domains such as A in Office-31 and Qdr in DomainNet lack sufficient domain information to support contrary attention. Nevertheless, our ADNT still has a top performance on these domains, which reflects the robustness of our method.

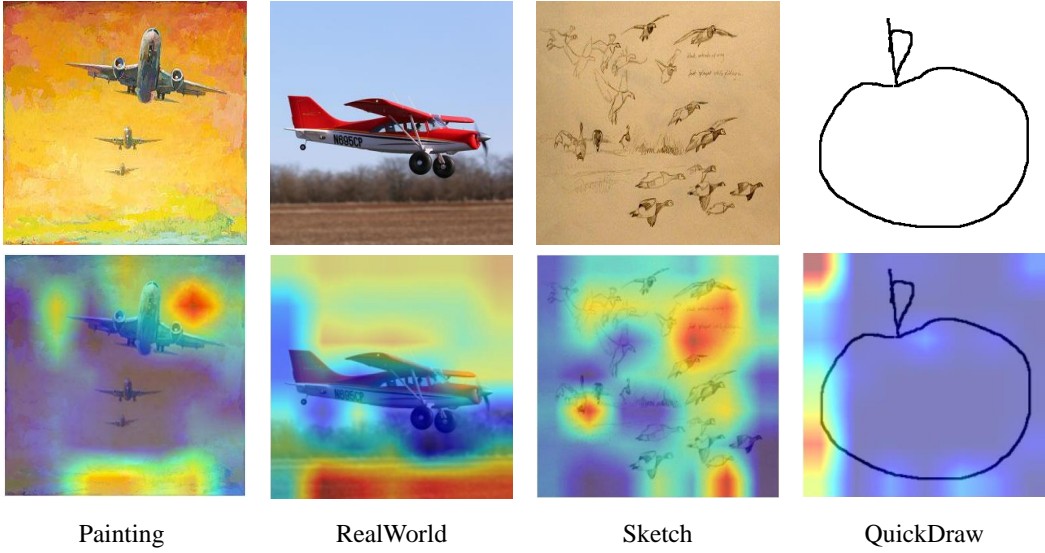

| Painting | RealWorld | Sketch | QuickDraw |

Figure 4: The visualization about Grad-CAM. The first row shows the original image and the second row shows the visualization.

Furthermore, the weight value in $A'$ actually reflects the correlation among domains (Here, we use the domain to refer to the features in the domain). Therefore, the relation of different domains can be estimated by $A'$. In this paper, we expect the following domain relationship: Initially, they should have very different styles, and the weights in $A'$ will be quite uncertain. Then, as the optimization of the model parameters, different domains will gradually be mixed. Finally, all domains should have a consistent style and maintain a balanced relationship. Therefore, the merging of all domains together means that there should be little difference among the final correlation coefficients in $A'$. So we calculate the standard deviation of $A'$ during the training.

As shown in the Fig. 5, during the training process, the standard deviation gradually becomes smaller and eventually tends to be stable, which indicates that the features from different domains are distributed in the space with a uniform state.

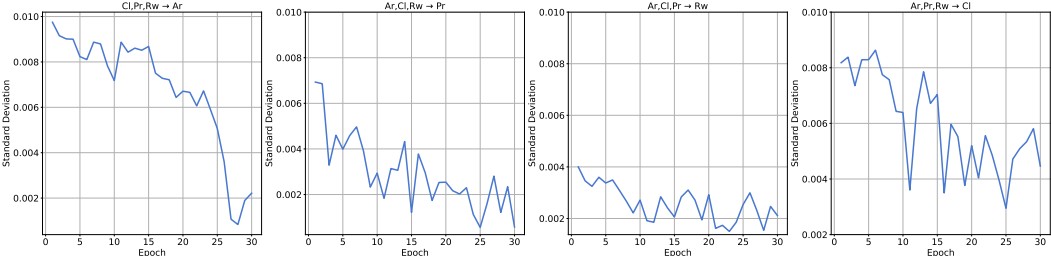

Figure 5: The standard deviation of $A'$ on the Office-Home dataset.

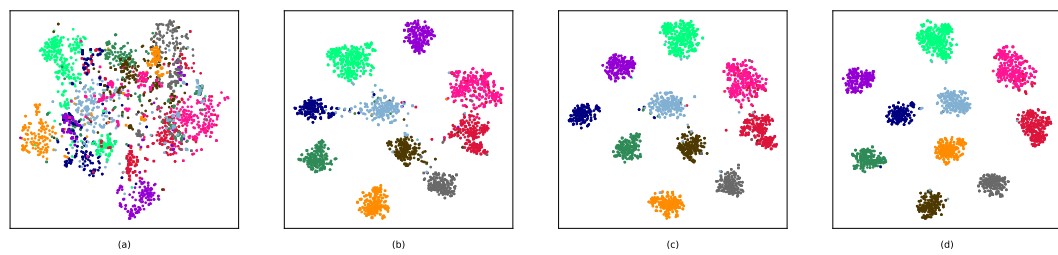

Figure 6: Visualization of the feature distribution during the training process in the transfer task Ar, Cl, Rw to Pr. (a), (b), (c) and (d) represent the epoch 1, 10, 20 and 30, respectively. To provide a clearer illustration, we select features from the first ten classes of each domain, with different color representing different class. Solid circles represent all source domains, while the squares represent target domains. Best viewed enlarged.

However, the above results alone do not demonstrate that features from different domains are deeply fused because the previous effect in Fig. 5 can also be achieved if features are sufficiently separated from each other. Consequently, we map the fused features **h** into the low-dimensional space and visualize the distribution during the training process, divided by different domains and categories. As shown in Fig. 6, at the start of training, the fused features are distributed haphazardly in space, and the network does not have capacity to process them. Then, different domains gradually move toward each other and merge together as the updating of model parameters. Finally, the features of the same category become more compact while preserving domain fusion. This powerfully demonstrates that through CADM, different domains embrace each other and achieve deep fusion. We make the model learn how to handle the fused features and show its excellent results.

## A.2  ADDITIONAL EXPERIMENTS ABOUT $\mathcal{L}_{AR-CE}$

First, as shown in the Fig. 7, we show the specific structure of the pseudo-label generation and the gradient back-propagation. Through $\mathcal{L}_{AR-CE}$, the soft label generation can be optimized during gradient descent to weaken the influence from the noise in the pseudo-labels.

Furthermore, the accuracy of the generated pseudo labels can reflect the effectiveness and stability of the pseudo-label generation. Therefore, we compare the average accuracy of pseudo labels generated with and without AR-CE loss on Office-31, Office-Caltech and Office-Home, which is shown in Fig. 7. Specifically, we calculate the mean accuracy of the final five epochs during training. It can be seen that after the introduction of the AR-CE loss, the model can derive noisy labels through self-correction and achieve the high accuracy of pseudo labels.

Finally, we have conducted additional experiments to vertify the effect of adaptive factors. Specifically, we counted 1000 samples with correct and wrong pseudo labels separately and calculated their distribution entropy as described in Sec. 4.3. The results are shown in Fig. in the appendix. It can be seen that the distribution entropy of the correct samples has values around 4, while the incorrect samples have smaller values, mostly around 2.5. This difference proves the feasibility of

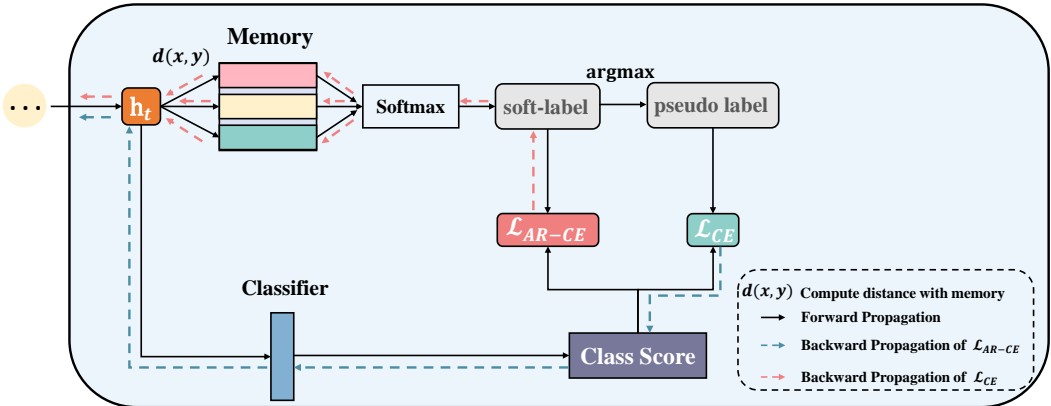

Figure 7: The specific structure of the pseudo-label generation in our ADNT, where the Class Score is the output of classifier.

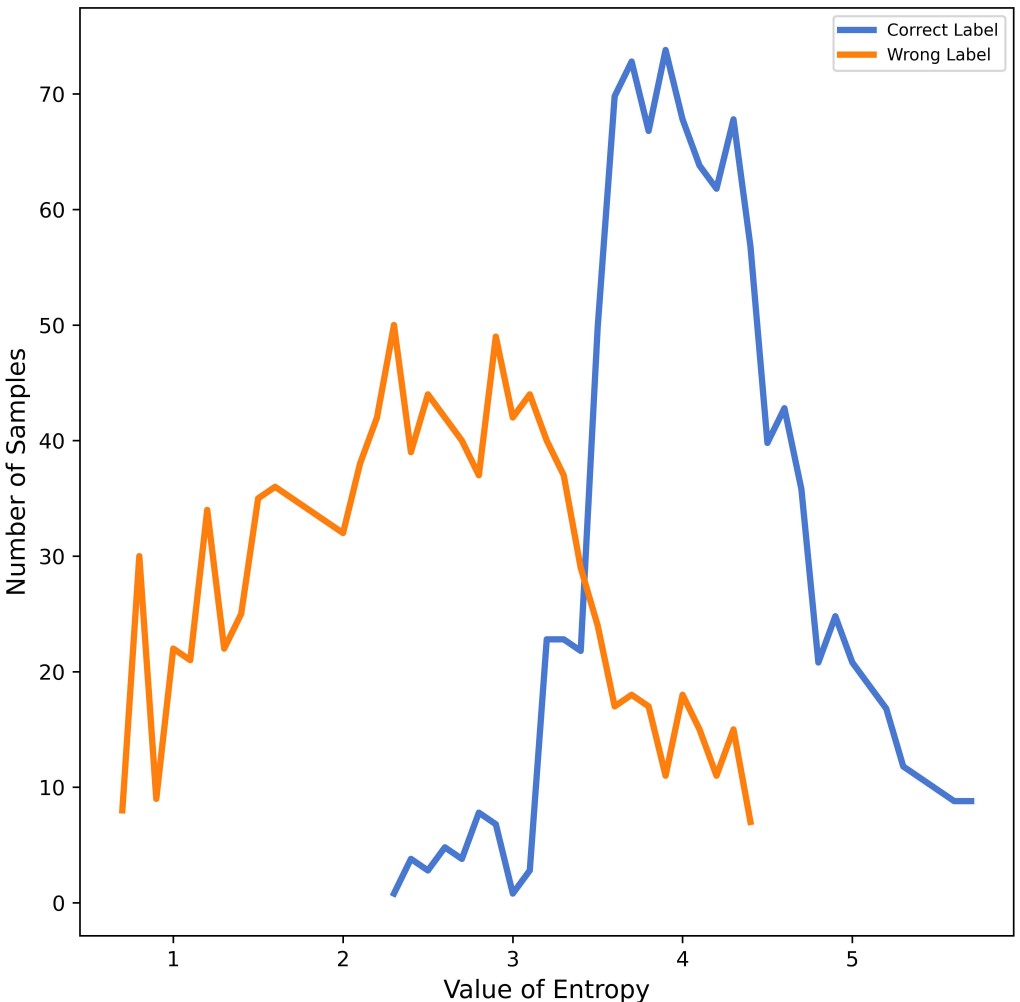

Figure 8: Visualization of the value of the distribution entropy for different samples, where the vertical coordinate is the number of samples and the horizontal coordinate is the value of the distribution entropy.

our proposed adaptive factor. Although it is difficult to be completely correct because the pseudo labels depend on the generation, our method can significantly achieve the correction of pseudo-label generation. As a result, the impact of noise on the model performance is reduced.

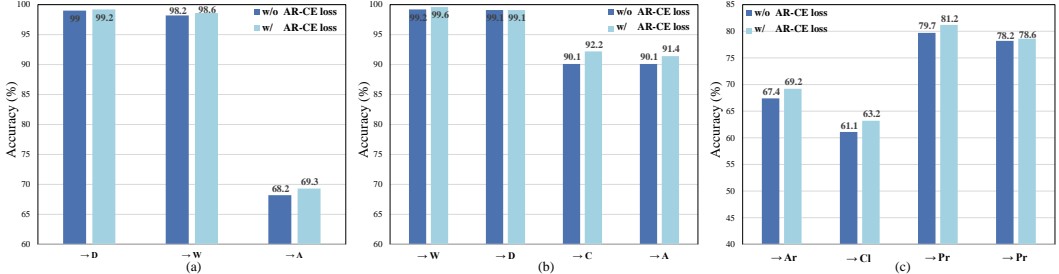

Figure 9: Mean accuracies (%) of pseudo labels on three datasets. (a), (b), (c) represent the result on Office-31, Office-Caltech and Office-Home respectively.

## A.3    MODEL ARCHITECTURE

Since the feature extractor $F()$ in our method is the pre-trained ResNet-50, in order to explore the performance of our method using a larger network structure, we conducted experiments on Office-31, Office-Caltech, and Office-Home using ResNet-101 as the feature extractor, as shown in Table 6, 7, 8. The results show that the performance of the network does improve somewhat, however, there is a drop in some tasks (especially some tasks that have close to 100% accuracy with ResNet-50). On the Office-Home dataset, there is a significant improvement. This indicates that data size is an important factor limiting the application of large model to our approach. Moreover, since we simply perform model replacement in this experiment, there is further room to optimize the way the larger model is combined with our proposed ADNT.

Table 6: Classification accuracies (%) on Office-31 dataset

| Method | A,W →D | A,D →W | D,W →A | Avg |
|---|---|---|---|---|
| ADNT (ResNet-50) | 100 | 99.6 | 74.4 | 91.3 |
| ADNT (ResNet-101) | 99.7 | 99.3 | 76.1 | 91.7 |

Table 7: Classification accuracies (%) on Office-Caltech dataset

| Method | A,C,D →W | A,C,W →D | A,W,D →C | C,D,W →A | Avg |
|---|---|---|---|---|---|
| ADNT (ResNet-50) | 100 | 100 | 97.6 | 96.3 | 98.5 |
| ADNT (ResNet-101) | 99.8 | 100 | 98.1 | 96.5 | 98.6 |

Table 8: Classification accuracies (%) on Office-Home dataset

| Method | Cl,Pr,Rw →Ar | Ar,Pr,Rw →Cl | Ar,Cl,Rw →Pr | Ar,Cl,Pr →Rw | Avg |
|---|---|---|---|---|---|
| ADNT (ResNet-50) | 73.8 | 66.5 | 85.1 | 83.3 | 77.2 |
| ADNT (ResNet-101) | 74.6 | 66.3 | 85.7 | 84.8 | 77.9 |

## A.4    PARAMETER ANALYSIS

Exponential moving average (EMA) is used in the experiments to maintain the domain style centers and class-level features, so the selection of hyperparameters $\alpha$ and $\beta$ requires additional analysis and

verification. Therefore, we perform an experimental comparison of their values and present them in Fig. 10.

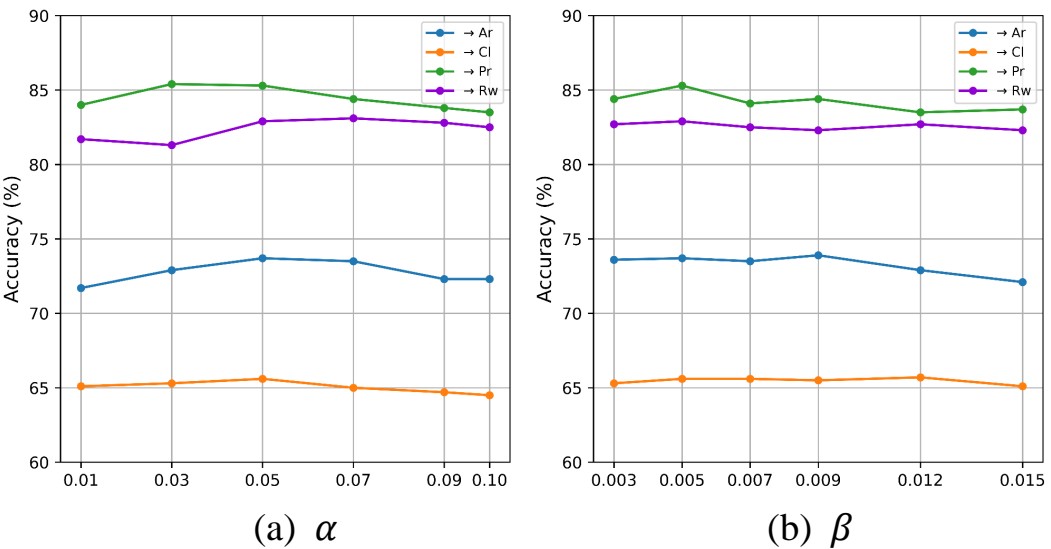

Figure 10: Sensitivity analysis of $\alpha$ and $\beta$.

As can be seen, the value of the hyperparameter $\alpha$ is somewhat greater and the $\beta$ is taken at an overall smaller value. This indicates that the domain style centroid is updated more vigorously and provides a gradient sufficient for the backpropagation of the network. Memory $M$, on the other hand, requires a slower update to maintain network robustness because it involves the pseudo-label generation.

