# OpenReview forum: "Joint Attention-Driven Domain Fusion and Noise-Tolerant Learning for Multi-Source Domain Adaptation"
_ICLR.cc/2023/Conference — Submitted to ICLR 2023_

### Official Review · Reviewer_2VJ6 · 2022-10-23

**Confidence:** 3
**Correctness:** 4
**Technical Novelty And Significance:** 3
**Empirical Novelty And Significance:** 2
**Recommendation:** 8

**Clarity, Quality, Novelty And Reproducibility:**

The writing of this article is clear, the structure and experiment design are reasonable, and it has a relatively high originality.

**Strength And Weaknesses:**

Strength:
1) The elaborately designed CADM module is inspired by the cross-attention mechanism. By adding a metric-based loss for maintaining a clear decision boundary and a memory M for storing features, the domain feature fusion capability of this module has been further improved. This fusion strategy may generalize to other visual MUDA tasks, like object detection and semantic segmentation.
2) The proposed new loss function AR-CE focuses on alleviating the influence of noisy pseudo-label generated from the target domain during training, which is inspired by the method SHOT++. By adding the reverse operation, the new loss can achieve pretty good results.
3) The presented method ADNT has obtained SoTA performances on four common MUDA benchmarks.

Weaknesses:
1) As mentioned above, the two main strategies involved in the method are derived from the improvement of existing modules. This limits the innovation and novelty of the paper to a certain extent.
2) The explanation of proposed loss AR-CE around Equations (15), (16) and (17) is ambiguous and difficult to understand. I think authors may need to rearrange the language to describe their design ideas.


**Summary Of The Paper:**

This paper proposed a novel method named ADNT for tackling the multi-source unsupervised domain adaptation (MUDA) problem. Their method mainly contains two well-designed strategies: the contrary attention-based domain merge (CADM) module for domain feature fusion, and the adaptive and reverse cross entropy (AR-CE) loss for robust pseudo-label generation. They verified the large superiority of ADNT on four MUDA benchmarks.

**Summary Of The Review:**

Although most of the two main strategies are inspired by other closely related researches, the method proposed in this paper is practical and effective, and can be used as a representative method in the field of MUDA.

---

> ### Author Response · Authors · 2022-11-16
> **Response to Reviewer 2VJ6**
>
> Thank you for your positive reviews of our paper. We respond to your two main concerns in the following:
>
> **Q1:** As mentioned above, the two main strategies involved in the method are derived from the improvement of existing modules. This limits the innovation and novelty of the paper to a certain extent.
>
> **A1:**
>
> ### (1). CADM
> First, we highlight and add to the motivation of our CADM in the third and fourth paragraphs of Sec. 1 and the first paragraph of Sec. 4.2, marked in blue. For a visual representation, we provide the following:
>
> Overall, since the main challenge of MUDA is to eliminate the differences between all domains, there are two main ways to achieve this. One is to extract domain invariant features among all domains, i.e., filter domain-specific information for different domains. The other is by mixing domain-specific information from different domains so that all domains share such mixed information and thus **fuse into one domain**. Previous approaches have mostly followed the former, however, filtering the domain-specific information for multiple domains can be difficult and often results in losing discrimination ability. For the latter, few methods have been proposed to address MUDA in this way, and there is a lack of effective frameworks to achieve such domain fusion, which is the main problem to be addressed by our proposed approach.
>
> Firstly, we propose a Contrary Attention-based Domain Merge (CADM) module, whose role is to perform domain fusion. Self attention can capture the higher-order correlation between features and emphases more relevant feature information, *e.g.,*, the semantically closest information.
> Differently, our CADM proposes the contrary attention, enabling each domain to pay more attention to semantically different domain-specific information of other domains.
> Then, by integrating these domain-specific information, each domain can achieve movement to other domains, thus resulting in domain fusion.
>
> The proposed CADM enables the passing of domain-specific information to achieve the deep fusion of all domains. For each domain, we expect it can receive domain-specific information from other domains by the information transfer to move towards other domains as shown in Fig. 1. To achieve this goal, the transfer of semantically distinct domain-specific information should be encouraged. It is because receiving semantically close domain-specific information primarily strengthens the original domain information and does not help eliminate the domain shift. Instead, receiving different domain-specific information can contribute to the mixing of domain-specific information and `push' the domain to other domains, thus achieving domain fusion.
>
> ### (2). $\mathcal{L}_{AR-CE}$
> For $\mathcal{L}_{AR-CE}$  ,  although borrowing ideas from reverse cross entropy loss, to the best of our knowledge, our method is the first to propose to assign gradients to pseudo-label generation. Therefore, our method can optimize the pseudo-label generation through the obtained soft labels.
> Furthermore, regarding your mention of being based on SHOT++ [1], in SHOT++ though they mention the phenomenon that when the model has a good distinguishing ability for one sample, the output of classifier should be close to the one-hot vector.
> However, they do not propose to conceptualize the discriminability of the network in terms of distribution entropy, which is first proposed by our approach. The experiments in ablation study and Sec. A.3 in Appendix can also reflect the effect of the adaptive factor.
>
> ---
>
> **Q2:** The explanation of the proposed loss AR-CE around Equations (15), (16), and (17) is ambiguous and difficult to understand. I think authors may need to rearrange the language to describe their design ideas.
>
> **A2:** Thank you for your feedback. **We have modified the presentation in Section 4.3, marked in blue.**
>
> Finally, thank you for reading, and we welcome any follow-up questions.

---

> > ### Comment · Reviewer_2VJ6 · 2022-12-11
> > **Thank you for your careful reply**
> >
> > Thank you for your careful reply. Now it looks like other reviewers are skeptical that the domain merge or fusion operation makes sense. If CADM can indeed play a great role in improving performance, I believe it can continue to be promoted in other more complex domain adaptation fields.

---

> > > ### Author Response · Authors · 2022-12-12
> > > **Response to Reviewer 2VJ6**
> > >
> > > Thank you for your feedback. For your concern about the meaning of domain fusion, we have detailed the motivation and meaning in the Introduction in the revision. In simple, domain fusion enables different domains to belong to the same one. Thus the core problem of multi-source domain adaptation, i.e., the differences among domains, can be well resolved. As a result, domain fusion is theoretically helpful for unsupervised multi-source domain adaptation, and the key lies in how this can be effectively achieved. To illustrate that our CADM can achieve this goal well and help the domain adaptation field, we demonstrate it in the following aspects.
> > >
> > > First, whether the method achieves the focus on domain-specific information as claimed: since our approach aims to achieve the domain fusion through the transfer of domain-specific information by contrary attention, we visualize the gradient of weights in the attention map by Grad-CAM in the Appendix. As illustrated in Figure 4, CADM can focus on domain-specific information, especially stylistic information such as texture, background, etc. This provides the basis for domain fusion by mixing different domain-specific information.
> > >
> > > The second is whether domain fusion can be achieved. On the one hand, we have visualized the data distribution by t-sne, and Figures 3, 6 demonstrate that CADM can eliminate the distribution differences among domains and achieve fusion. On the other hand, the visualization of the variance in Figure 5 proves that different domain distributions change in the way we expect and eventually converge.
> > >
> > > The final issue is whether this domain fusion leads to better performance. Based on the ablation experiments results, it can be seen that CADM has a significant performance improvement, and the comparison with the ADM method can further demonstrate the role of our proposed contrary attention.

---

### Official Review · Reviewer_buhi · 2022-10-24

**Confidence:** 4
**Correctness:** 3
**Technical Novelty And Significance:** 2
**Empirical Novelty And Significance:** 2
**Recommendation:** 3

**Clarity, Quality, Novelty And Reproducibility:**

The motivation is not clearly explained. Why domain-specific information is important in domain adaptation is not well analyzed. This results in poor clarity.
The quality is generally good, especially the CADM module and the proposed new cross-entropy loss function.
Some important implementation details are missing, and the code is not provided. I doubt the reproducibility of this paper.

**Strength And Weaknesses:**

Strengths:
1. The idea of transfering domain-specific information is interesting though I am not sure whether this makes sense.
2. The proposed CADM and adaptive and reverse cross-entropy loss are relatively novel based on some existing techniques.
3. The results on some adaptation settings are good and the ablation studies and visualizations are convincing.

Weaknesses:
1. The motivation is not clearly explained. Learning domain-invariant features and enhancing the category information are easy to understand for multi-source domain adaptation. However, this paper tries to transfer domain-specific information. Since they are domain-specific, how can they be transferred and why do they work for adaptation?
2. As some methods claimed, different source domains and different examples in each source domain play different roles during the adaptation process. How is such information considered?
3. It seems that one feature extractor F and one classifer C are firstly pre-trained. Since there are multiple domains, do you think simply combining all source samples and train one common F and C makes sense?
4. The results on DomainNet are much better than the compared baselines, but the result on other three datasets are only marginally better or even inferior. Why? There is no analysis on the conditions that the proposed method work.
5. The presentation needs to be revised and improved. For example, what does $D$ indicate (feature dimension in my understanding)? In Introduction, "Secondly, To enable"->"Secondly, to enable". The format of references, especially the names of conferences and journals, is inconsistent.


**Summary Of The Paper:**

This paper studies multi-source domain adaptation, an interesting and important domain adaptation topic. Instead of learning domain-invariant feature representations, this paper proposes to transfer domain-specific information and also considers pseudo-label generation. Specifically, this paper proposes (1) a Contrary Attention-based Domain Merge (CADM) module to enable the interaction among the features so that domain-specific information can be mixed, and (2) an adaptive and reverse cross-entropy loss to correct the pseudo labels during training. Experiments are conducted on four benchmark datasets for classification.

**Summary Of The Review:**

Interesting idea but unclear motivation, generally new components for multi-source domain adaptation, insufficient analysis on the results, and could be improved presentation.

---

> ### Author Response · Authors · 2022-11-16
> **Response to Reviewer buhi (1/2)**
>
> Thank you for your reviews of our paper. For your doubts about reproducibility, the main code of our approach has been uploaded to the supplementary material.
> Then we respond to your questions in the following:
>
> **Q1:** The motivation is not clearly explained.
>
> **A1:** Thanks for your comments. Learning domain-invariant features and enhancing the category information are easy to understand for multi-source domain adaptation. However, as we stated in the main text, extracting domain-invariant features is more difficult when the number of domains is larger. We have conducted experiments to verify this conclusion, referring to the **Response to Reviewer eMFp about question 2**.  Our method is more robust when facing the number of domain increase than the method of extracting domain invariant features.
>
> Furthermore, the goal of our approach is for each domain to move to other domains by receiving domain-specific information from other domains, and eventually achieving domain fusion.
> This is the role of domain-specific information in MUDA, i.e., the mixing of domain-specific information is achieved in the above way, so that all domains belong to the same domain and differences between domains are eliminated. This is the difference between our ADNT and previous approaches, the previous methods focus on extracting domain invariant features, while our approach aims at domain fusion.
> For why domain-specific information can be transferred. Specifically, the transfer of domain-specific information is achieved through the Contrary Attention map in CADM, which focuses more on the distinction between domains and enables the transfer of domain-specific information based on this distinction.
>
> **A more detailed explanation of the above motivation and CADM is added and modified in the third and fourth paragraphs of Sec. 1 and the first paragraph of Sec. 4.2, marked in blue.**
>
> In addition, to demonstrate that our approach does achieve focus and transfer of domain-specific information, **we use Grad-CAM [1] to perform the visualization of contrary attention in Sec. A.1 in Appendix.**
> The specific images and analysis are shown in Fig. 4 and the first paragraph of the Appendix. This visualization can demonstrate that our approach can focus more on other domains with domain-specific information such as texture, background, etc.
>
> ---
>
> **Q2:** As some methods claimed, different source domains and different examples in each source domain play different roles during the adaptation process. How is such information considered?
>
> **A2:** The adaptation is actually the process of joint training in the source domain data and the unlabeled target domain data using the model pre-trained in the source domain. In this process, since we expect to achieve the fusion of all domains, we construct the correlation of different domains with samples from different domains by contrary attention. Therefore, the role of different domains and samples is based on the relationship captured by contrary attention.
>
> ---
>
> **Q3:** It seems that one feature extractor F and one classifer C are firstly pre-trained. Since there are multiple domains, do you think simply combining all source samples and train one common F and C makes sense?
>
> **A3:** As presented in the paper, the goal of our approach is to achieve domain fusion so that all domains are treated as coming from the same one. However, using different $F$ and $C$ for each domain actually causes the model to learn different distributions for different domains. Specifically, during the learning process, the particular $F$ and $C$ are updated according to the features of the particular domain, which can lead to differences in the distribution among domains. This may be effective for extracting domain-invariant features, but is not meaningful for domain fusion. As a result, we use one $F$ and $C$ for all domains.

---

> ### Author Response · Authors · 2022-11-17
> **Response to Reviewer buhi (2/2)**
>
> ---
>
> **Q4:** The results on DomainNet are much better than the compared baselines, but the result on other three datasets are only marginally better or even inferior. Why? There is no analysis on the conditions that the proposed method work.
>
> **A4:** The reason why our method is only $0.1\%$ higher than the best DINE method on Office-home is that, as explained in the notes of the Table 3, DINE is based on the ResNet-101, while our ADNT is based on ResNet-50. Compared to those ResNet-50-based methods, ADNT has a $0.9\%$ improvement, which is significant in this dataset.
> On the Office-caltech dataset, our method is $0.2\%$ higher than the state-of-the-art algorithms. Since the accuracy on this dataset is already close to $99\%$, the $0.2\%$ improvement is difficult, and previous methods, such as PTMDA [2], DECISION [3], SHOT++ [4], and so on, also bring the same degree of improvement.
> On the Office-31 dataset, our method achieves optimal results on two tasks, but is inferior than the optimal method overall.
> For the performance degradation tasks, we perform the analysis in Sec. 5.2 and Appendix. In addition, for comparison with the ResNet-101-based method, we conducted experiments with ResNet-101 as the feature extractor, as detailed in Appendix Sec. A.3.
>
> ---
>
> **Q5:** The presentation needs to be revised and improved.
>
> **A5:** Thank you for the suggestions, we have revised the issues you mentioned in the revision.
>
> Finally, thank you for reading and we welcome any follow-up questions.
>
> ---
>
> ### Refs
> [1] R. R. Selvaraju, M. Cogswell, A. Das, R. Vedantam, D. Parikh and D. Batra, "Grad-CAM: Visual Explanations from Deep Networks via Gradient-Based Localization," 2017 IEEE International Conference on Computer Vision (ICCV), 2017, pp. 618-626, doi: 10.1109/ICCV.2017.74.

---

### Official Review · Reviewer_eMFp · 2022-11-04

**Confidence:** 3
**Correctness:** 2
**Technical Novelty And Significance:** 3
**Empirical Novelty And Significance:** 3
**Recommendation:** 5

**Clarity, Quality, Novelty And Reproducibility:**

Even though the proposed approach seems novel in the research area, the novelty is not supported by well-defined motivation. The lack of motivation makes the designed framework unreasonable, which makes it hard to understand the reason why the performance could increase through the proposed algorithm.

**Strength And Weaknesses:**

The visualization figures of the paper are helpful to understand the overall framework, and the experimental settings are reasonable to validate the algorithm. The state-of-the-art performance of the proposed algorithm confirms its validity, and several ablation studies verify the effectiveness of each component.

However, I have several concerns and questions yet.

1. Weak motivation for sharing the same domain-specific information

   The authors insisted that filtering the domain-specific information for every domain is difficult and often results in losing discrimination ability. Of course, the filtering process can be challenging, but the approach looks more reasonable than the shared domain-specific information. Since the multiple domains would have different characteristics that are helpful to extend the distinguishability of the target domain, it seems more reasonable to extract the different domain-specific information from each domain. In addition, when we want to extract the shared domain-specific information from the multi-source, only the limited information would be extracted when the multiple sources are composed of dispersed domains. I hope to know the additional explanation for this problem.

2. The organization of sampled batch

 In the first paragraph of section 3, the sampled batch contains "b" samples for each domain. Then, when we have a variety of source domains, the number of domain-wise samples becomes very limited. I hope to know the relationship between the number of domains and the performance of the proposed algorithm.

3. Meaning of Eq.9

  The training loss of Eq.9 seems contradictive with Eq.5. While Eq.5 makes the similar domain information connected less effectively, Eq.9 lets the scattered information be gathered again. Why do we need to consider the two contradictive losses in the integrated form? And what would be the effectiveness of the integration?

4. Missing explanation of Table1, 2, 3

  There is no explanation for Tables 1, 2, and 3. Since the results from Tables 1, 2, and 3 are from the main experiments, the related analysis is essential but missing.

5. MIssing analysis for the performance drop

In Tables 3 and 4, there appear several performance drops compared to the state-of-the-art algorithms for some scenarios, but no related analysis is explained in the paper. We can catch the effectiveness of the proposed algorithm through the analysis of limitations.

6. Several minor comments

 In Figure 4, the font size is too small
At the last of Eq. 6 and 7, commas should be followed to continue the sentences.

**Summary Of The Paper:**

This paper targets transferring multi-source knowledge to the target domain. While the previous studies tried to utilize the generalized knowledge of the multiple source domains, the proposed algorithm focus on domain-specific information. To accomplish the extraction of the domain-specific information, the paper presents new modules of a contrary attention-based domain merge module and an adaptive and reverse cross-entropy loss. The contrary attention-based domain merge module can handle the interaction among the features for the domain-specific information. In addition, the adaptive and reverse cross-entropy loss works as a constraint for stable pseudo-label generation. The overall framework shows the state-of-the-art performance in the various experimental scenarios.

**Summary Of The Review:**

Due to the weakly explained motivation of the proposed algorithm and the missing analysis of the performance drop, it becomes hard to understand the reasoning of the framework design and its state-of-the-art performance. In addition, the overall organization of the paper should be improved and proofread. The details can be referred to the weakness section.

---

> ### Author Response · Authors · 2022-11-16
> **Response to Reviewer eMFp (1/2)**
>
> Thank you for your reviews of our paper. We respond to your questions in the following:
>
> **Q1:** Weak motivation for sharing the same domain-specific information
>
> **A1:** Thanks for your feedback. Our approach is not to extract the shared domain-specific information, but to mix domain-specific information from all domains. This mixing makes all domains belong to the same domain, thus achieving domain fusion. **Since our previous expression may have caused a misunderstanding, we have revised and added relevant contents in the revisions to clarify the motivation and rationale of our approach, specifically in the third and fourth paragraphs of Sec. 1 and the first paragraph of Sec. 4.2, marked in blue.** Some of the explanations are as follows:
>
> Overall, since the main challenge of MUDA is to eliminate the differences between all domains, there are two main ways to achieve this. One is to extract domain invariant features among all domains, i.e., filter domain-specific information for different domains. The other is by mixing domain-specific information from different domains so that all domains share such mixed information and thus **fuse into one domain**. Previous approaches have mostly followed the former, however, filtering the domain-specific information for multiple domains can be difficult and often results in losing discrimination ability. For the latter, few methods have been proposed to address MUDA in this way, and there is a lack of effective frameworks to achieve such domain fusion, which is the main problem to be addressed by our proposed approach.
>
> Firstly, we propose a Contrary Attention-based Domain Merge (CADM) module, whose role is to perform the domain fusion. Self attention can capture the higher-order correlation between features and emphases more relevant feature information, *e.g.*, the semantically closest information.
> Differently, our CADM proposes the contrary attention, enabling each domain to pay more attention to semantically different domain-specific information of other domains.
> Then, by integrating these domain-specific information, each domain can achieve movement to other domains, thus resulting in domain fusion.
>
> The proposed CADM enables the passing of domain-specific information to achieve the deep fusion of all domains. For each domain, we expect it can receive domain-specific information from other domains by the information transfer to move towards other domains as shown in Fig. 1. To achieve this goal, the transfer of semantically distinct domain-specific information should be encouraged. It is because receiving semantically close domain-specific information primarily strengthens the original domain information and does not help eliminate the domain shift. Instead, receiving different domain-specific information can contribute to the mixing of domain-specific information and `push' the domain to other domains, thus achieving domain fusion.
>
> ---
>
> **Q2:** The organization of sampled batch
>
> **A2:** Thanks for your suggestions. Notably, the number of domains in the current MUDA dataset is $3-6$, so there is no such phenomenon as too many domains leading to too few samples. Furthermore, exploring the relationship between the number of domains and network performance makes sense. However, directly decreasing or increasing the number of domains does not satisfy the control variables because simply changing the number of domains changes the size of the training data, so the direct comparison does not correctly reflect the relation between the performance of the network and the number of domains. Therefore, we first fixed the total number of samples and conducted experiments on Office-home dataset, and the experimental results will be given soon.
>
> ---
>
> **Q3:** Meaning of Eq.9
>
> **A3:** As introduced in Sec. 4.2, Eq. 5-8 are designed to allow domain to receive domain-specific information from other domains and thus enable movement to other domains. Eq. 9 aims to guarantee the discriminating ability of model by maintaining the intra-class compactness during the domain movement. Specifically, Eq. 9 achieves the above goal by making the features of the same class close enough in feature space. As a result, one focuses on the fusion of all domains, and one focuses on the intra-class compactness. These two equations do not conflict, but can achieve a trade-off between domain fusion and good decision boundaries.

---

> > ### Author Response · Authors · 2022-11-16
> > **Response to Reviewer eMFp (2/2)**
> >
> > **Q4:** Missing explanation of Table 1, 2, 3. There is no explanation for Tables 1, 2, and 3. Since the results from Tables 1, 2, and 3 are from the main experiments, the related analysis is essential but missing.
> >
> > **A4:** Thank you for your feedback. We add related analysis of the main experimental results in the revision, as detailed in Sec 5.2.
> >
> > ---
> >
> > **Q5:** MIssing analysis for the performance drop
> >
> > **A5:** In MUDA, no method can achieve the best results on all tasks because different methods have different focuses and different advantages. Similarly, even though our methods achieve the best on most tasks, performance degradation compared to previous methods appears. Consequently, we have analyzed the performance degradation and added it in Sec. 4 of the main text and the Sec. A.2 of Appendix. In general, one reason for the performance degradation is the difference in model structure, i.e., the previous method uses ResNet-101 while our method uses ResNet-50. in addition, another main reason is that our method does not achieve the best performance when dealing with target domains with less domain information, as detailed in Fig. 4 in the Appendix and the related explanation.
> >
> > ---
> >
> > **Q6:** Several minor comments
> >
> > **A6:** We appreciate your feedback. We have made modifications in the revision as suggested, and in addition, due to space limitations, the figure has been placed in the Appendix.

---

> ### Author Response · Authors · 2022-11-17
> **Response to Reviewer eMFp about question 2**
>
> Here we present the experimental results concerning the question 2. As previously stated, simply increasing or decreasing the domain would lead to an unreasonable comparison due to the difference in data size. Therefore, we fix the number of samples in the source domain to 3000, i.e., 3000 training samples for a single source domain, 1500 + 1500 for two, and 1000 * 3 for three. We conducted experiments on the Office-home dataset for transfer to Ar and Pr respectively, and compare it with the $\text {SImpAl}_{50}$ of extracting domain invariant features. The results are as follows:
>
> | Method            | Cl $\rightarrow$Ar | Cl,Pr$\rightarrow$Ar |  Cl,Pr,Rw $\rightarrow$Ar |
> |:-------------------:|:--------------------------:|:--------------------------:|:--------------------------:|
> | $\text {SImpAl}_{50}$  | 60.0                     | 66.7                     | 64.1                     |
> | ADNT   | 57.3                     | 68.7                     | 68.1                     |
>
> | Method            | Rw $\rightarrow$Pr| Rw,Cl$\rightarrow$Pr|  Rw,Cl,Ar $\rightarrow$Pr|
> |:-------------------:|:--------------------------:|:--------------------------:|:--------------------------:|
> | $\text {SImpAl}_{50}$  | 71.2                     | 75.3                     | 72.1                     |
> | ADNT   | 74.3                     | 80.7                     | 80.2                     |
>
> It can be seen that for the same data size, the relation between the number of source domains and performance is $2 > 3 > 1$. This implies that although MUDA outperforms SUDA, the MUDA task is more challenging when the number of source domains increases. Moreover, the results of $\text {SImpAl}_{50}$ using three source domains show a significant decrease compared to two, which can confirm what we have stated in the paper that it is more challenging to extract domain invariant features when the number of domains is larger. In contrast, our method proposes to perform domain fusion instead of extracting domain-invariant features, and thus has more stable performance in the face of the increasing number of source domains.

---

### Official Review · Reviewer_javN · 2022-11-05

**Confidence:** 3
**Correctness:** 4
**Technical Novelty And Significance:** 2
**Empirical Novelty And Significance:** 2
**Recommendation:** 5

**Clarity, Quality, Novelty And Reproducibility:**

The paper is clear. However, the idea is not good enough, it is like leveraging current information for domain fusion.

**Strength And Weaknesses:**

1. Figure 1 doesn't convey the pipeline and process of your model well, especially in the right part concerning the classifier and pseudo-label generation. Adding some sub-figures may be better.
2. As you proposed to assign gradients to the pseudo label generation process, give a comparison of the traditional gradient-free pseudo generation process and your process in the form of the figure may be better. Or, you may point out the difference by giving annotations of the
arrows and lines in Figure 1.
3. The adaptive and reverse cross-entropy loss depends on the correctness of generated hard labels. However, there is no guarantee that the hard labels are reliable. The adaptive factor based on the entropy of distribution may amplify the influence of the wrong assignment and mislead the model. Have you ever considered about this and what about your explanation?
4. Is there any evidence to validate the contribution of reverse cross-entropy in the ablation study? This may validate the effectiveness of the adaptive factor.
5. The annotations and explanations should be reconsidered. For example, if the adaptive and reverse cross-entropy loss is not added, does the method assign gradients to the pseudo-label generation process? If the gradients are assigned, what is the loss? reverse cross-entropy loss or others? This should be annotated clearly and added an explanation in the ablation study part.
6. The ResNet50-based models achieve 100% on some experiments on Office-31, Office-Caltech dataset outperformed the methods which are based on ResNet101. What are the results if your use ResNet101 on ADNT on the Office-31, Office-Caltech, and Office-Home
dataset?

**Summary Of The Paper:**

This work concentrates on two problems in the MUDA task: a) sharing the same domain-specific information across domains to fuse multi-source data; b) generating reliable pseudo labels to alleviate the effect of noisy pseudo labels. Thus, they proposed the Contrary Attention-based Domain Merge modules to pass domain-specific information through EMA-updated domain style centroids. Moreover, a growing class center is applied to optimize intra-class distance. In order to generate reliable pseudo labels, gradients are assigned to the generation process of pseudo labels. Additionally, adaptive and reverse cross-entropy loss is proposed to ensure reliable pseudo-label generation. Extensive experiments were conducted to validate their contributions.

**Summary Of The Review:**

The paper is a borderline paper for this conference.

---

> ### Author Response · Authors · 2022-11-16
> **Response to Reviewer javN (1/2)**
>
> Thank you for your reviews of our paper. We respond to your questions in the following:
>
> **Q1:** Figure 1 doesn't convey the pipeline and process of your model well, especially in the right part concerning the classifier and pseudo-label generation. Adding some sub-figures may be better.
>
> **A1:** Thank you for your feedback. For space limitation, we show the specific structure about pseudo-label generation and gradient backpropagation in Appendix (Fig.7 and Sec. A.2).
>
> ---
> **Q2:** As you proposed to assign gradients to the pseudo label generation process, give a comparison of the traditional gradient-free pseudo generation process and your process in the form of the figure may be better. Or, you may point out the difference by giving annotations of the arrows and lines in Figure 1.
>
> **A2:** Thank you for your feedback. Referring to A1, the specific structure of pseudo-label generation is added in Sec. A.2 in Appendix. General pseudo-label generation methods of MUDA can be referred to [1]. Specifically, they use the following approach to generate pseudo labels in the unlabeled target dataset: Firstly, they clustered the features in a fixed training stage (*e.g.*, epoch 5, 10) according to the network classification results and calculated the cluster centroids. Secondly, the distance between the features and the cluster centroids is computed. Finally, the pseudo-label of sample will be set as the class whose cluster centroid is closest to it.
>
> The difference between our approach and theirs is that, on the one hand, our approach can obtain real-time soft labels through the dynamic memory, and on the other hand, the $\mathcal{L}_{AR-CE} $  based on the soft labels allows the optimization of pseudo-label generation.
>
> ---
>
> **Q3:** The adaptive and reverse cross-entropy loss depends on the correctness of generated hard labels. However, there is no guarantee that the hard labels are reliable. The adaptive factor based on the entropy of distribution may amplify the influence of the wrong assignment and mislead the model. Have you ever considered about this and what about your explanation?
>
> **A3:** In fact, the purpose of reverse cross entropy loss is to weaken the impact caused by incorrect hard labels, and the adaptive factor provides adaptive adjustment for this weakening. **As the previous statement may cause misunderstanding, we have modified the presentation in Section 4.3, which is related to Questions 2, 3, and 5, and can be seen specifically in the text and formulas between Eq. 14 and Eq. 18.**
>
> In addition, we have conducted additional experiments on this issue of your concern that adaptive factors amplify incorrect assignments. We counted 1000 samples with correct and wrong pseudo labels separately and calculated their distribution entropy as described in Sec. 4.3. **The results are shown in the Appendix (Fig. 8 and the Sec. A.2).** It can be seen that the distribution entropy of the correct samples has values around $4$, while the incorrect samples have smaller values, mostly around $2.5$. This difference proves the feasibility of our proposed adaptive factor. Although it is difficult to be completely correct because the pseudo labels depend on the generation, our method can significantly achieve the correction of pseudo-label generation. As a result, the impact of noise on the model performance is reduced.
>
> ---
> **Q4:** Is there any evidence to validate the contribution of reverse cross-entropy in the ablation study? This may validate the effectiveness of the adaptive factor.
>
> **A4:** It's a good question, we supplement the ablation experiments about the reverse cross entropy loss in Table 5 in Sec. 5.3, denoted as $+ \mathcal{L}_{R-CE}$. For visual representation, we provide the experimental results below:
> | Method                  | Cl,Pr,Rw $\rightarrow$ Ar | Ar,Pr,Rw $\rightarrow$ Cl | Ar,Cl,Rw $\rightarrow$ Pr | Ar,Cl,Pr $\rightarrow$ Rw | Avg  |
> |:-------------------------:|:---------------------------:|:---------------------------:|:---------------------------:|:---------------------------:|:------:|
> | Baseline                | 67.1                      | 56.6                      | 78.6                      | 77.0                      | 69.8 |
> | + $\mathcal{L}_{R-CE}$  | 67.5                      | 62.9                      | 78.9                      | 77.4                      | 71.6 |
> | + $\mathcal{L}_{AR-CE}$ | 69.2                      | 63.1                      | 81.2                      | 78.6                      | 73.0 |
>
> As shown in above table, $\mathcal{L}_{R-CE}$ brings the performance improvement, and the adaptive factor is able to amplify this improvement. This demonstrates the positive and effective role of adaptive factor.

---

> > ### Author Response · Authors · 2022-11-16
> > **Response to Reviewer javN (2/2)**
> >
> > **Q5:** The annotations and explanations should be reconsidered. For example, if the adaptive and reverse cross-entropy loss is not added, does the method assign gradients to the pseudo-label generation process? If the gradients are assigned, what is the loss? reverse cross-entropy loss or others? This should be annotated clearly and added an explanation in the ablation study part.
> >
> > **A5:** For the loss functions used in our method, where only $\mathcal{L}_{AR-CE}\ $ assigns gradients to the pseudo-label generation,
> >
> > because only $\mathcal{L}_{AR-CE}\ $   involves the soft labels (others use hard label obtained by $\mathrm{argmax}$), which ensures the back-propagation of the gradients of labels.
> >
> > ---
> >
> > **Q6:** The ResNet50-based models achieve $100% $ on some experiments on Office-31, Office-Caltech dataset outperformed the methods which are based on ResNet101. What are the results if your use ResNet101 on ADNT on the Office-31, Office-Caltech, and Office-Home dataset?
> >
> > **A6:** **We add experiments using ResNet-101 as the feature extractor in Sec. A.3 and Table 6, 7, and 8 in Appendix.** For visualization, the results are shown below:
> > | Method            | A,W $\rightarrow$D | A,D $\rightarrow$W | D,W $\rightarrow$A | Avg  |
> > |:-------------------:|:--------------------:|:--------------------:|:--------------------:|:------:|
> > | ADNT (ResNet-50)  | 100                | 99.6               | 74.4               | 91.3 |
> > | ADNT (ResNet-101) | 99.7               | 99.3               | 76.1               | 91.7 |
> >
> > | Method            | A,C,D $\rightarrow$W | A,C,W $\rightarrow$D | A,W,D $\rightarrow$C | C,D,W $\rightarrow$A | Avg  |
> > |:-------------------:|:----------------------:|:----------------------:|:----------------------:|:----------------------:|:------:|
> > | ADNT (ResNet-50)  | 100                  | 100                  | 97.6                 | 96.3                 | 98.5 |
> > | ADNT (ResNet-101) | 99.8                 | 100                  | 98.1                 | 96.5                 | 98.6 |
> >
> > | Method            | Cl,Pr,Rw $\rightarrow$Ar | Ar,Pr,Rw $\rightarrow$Cl | Ar,Cl,Rw $\rightarrow$Pr | Ar,Cl,Pr $\rightarrow$Rw | Avg  |
> > |:-------------------:|:--------------------------:|:--------------------------:|:--------------------------:|:--------------------------:|:------:|
> > | ADNT (ResNet-50)  | 73.8                     | 66.5                     | 85.1                     | 83.3                     | 77.2 |
> > | ADNT (ResNet-101) | 74.6                     | 66.3                     | 85.7                     | 84.8                     | 77.9 |
> >
> >
> > The results show that the performance of the network does improve somewhat, however, there is a drop in some tasks (especially some tasks that have close to $100\%$ accuracy with ResNet-50).
> > On the Office-Home dataset, there is a significant improvement. This indicates that data size is an important factor limiting the application of large model to our approach. Moreover, since we simply perform model replacement in this experiment, there is further room to optimize the way the larger model is combined with our proposed ADNT.
> >
> > ---
> >
> > Finally, thank you for reading and we welcome any follow-up questions.
> >
> > ### Refs:
> >
> > [1]: Ahmed S M, Raychaudhuri D S, Paul S, et al. Unsupervised multi-source domain adaptation without access to source data[C]//Proceedings of the IEEE/CVF Conference on Computer Vision and Pattern Recognition. 2021: 10103-10112.

---

### Author Response · Authors · 2022-11-16
**Paper Revision**

We thank all the reviewers for their insightful feedback and suggestions for improving the paper. We are glad that the reviewers found our paper clear, practical, effective, has insightful idea, reasonable experimental settings, and convincing visualizations.

Below we summarize the updates in our revision:

**Introduction:** Since we found in some reviews that the motivation and innovation of the method was not well represented. Therefore, we made a new statement and modification of the motivation of the method in Introduction, in the third and fourth paragraphs.

**Sec. 4.2:** The principle and motivation of CADM are added in the first paragraph.

**Sec. 4.3:** To describe our proposed $\mathcal{L}_{AR-CE}\ $ more clearly, we have added and modified presentation.

**Sec. 5.2:** Based on the comments, we have added the analysis of the experiments and added ablation experiments.

**Appendix:** In the appendix, we add additional experimental analyses, including Grad-CAM visualization, the figure about the specific structure of the pseudo-label generation, visualization of the distribution entropy (adaptive factor), and experiments using ResNet-101 as the feature extractor. **Note that Fig 9 in the first paragraph of the appendix should be Fig.4. We apologize for this error.**

**To make it easier to review the updates to the paper, major updates have been made in blue text.**

**The main code of our method has been submitted to the supplementary material**

---

### Decision · Program_Chairs · 2023-01-20

**Decision:**

Reject

**Justification For Why Not Higher Score:**

Both the motivation of this new research direction and the effectiveness of the proposed method need to be further improved.

**Justification For Why Not Lower Score:**

N/A

**Metareview: Summary, Strengths And Weaknesses:**

This work adopts the less researched domain fusion framework by mixing domain-specific information for the MUDA problem. Research in this new direction rather than following the popular way by extracting and keeping domain invariant features among all domains is inspiring, although the relevant explanations are not detailed and convincing enough. One major concern is the improvement of the method seems not to be stable. For example, the results on DomainNet are much better than the compared baselines, but the result on other three datasets are only marginally better or even inferior. Although the authors explain this effect as the network architecture, e.g., ResNet-50 vs. ResNet 101, but PTMDA, which uses the same architecture as the proposed method, achieves similar performance on Office-31 dataset and OfficeCaltech dataset. It would make the paper more solid if the authors can clearly explain the effectivenss of the proposed on different datasets.